# Analysis of Metabolite Accumulation Related to Pod Color Variation of *Caragana intermedia*

**DOI:** 10.3390/molecules24040717

**Published:** 2019-02-16

**Authors:** Feiyun Yang, Tianrui Yang, Kun Liu, Qi Yang, Yongqing Wan, Ruigang Wang, Guojing Li

**Affiliations:** 1College of Life Sciences, Inner Mongolia Agricultural University, Inner Mongolia Key Laboratory of Plant Stress Physiology and Molecular Biology, Hohhot 010018, China; yangfeiyun@imau.edu.cn (F.Y.); yang_tianrui@163.com (T.Y.); liukun261@163.com (K.L.); atp_yangqi@163.com (Q.Y.); livgreen176@163.com (Y.W.); ruigangwang@126.com (R.W.); 2College of Food Science and Engineering, Inner Mongolia Agricultural University, Hohhot 010018, China

**Keywords:** pod color, *Caragana intermedia*, metabolomics, flavonoids

## Abstract

*Caragana intermedia*, a leguminous shrub widely distributed in cold and arid regions, is rich in secondary metabolites and natural active substances, with high nutritional and medical values. It is interesting that the pods of *C. intermedia* often show different colors among individual plants. In this study, 10-, 20- and 30-day-old red and green pods of *C. intermedia* were used to identify and characterize important metabolites associated with pod color. A total 557 metabolites, which could be classified into 21 groups, were detected in the pod extracts using liquid chromatography coupled with ESI-triple quadrupole-linear ion trap mass spectrometer (LC-ESI-MS/MS). Metabolomics analysis revealed significant differences in 15 groups of metabolites between red and green pods, including amino acids, nucleotide derivatives, flavonoids, and phytohormones. Metabolic pathway analysis showed that the shikimic acid and the phytohormone metabolic pathways were extraordinarily active in red pods, and the difference between red and green pods was obvious. Moreover, red pods showed remarkable flavonoids, cytokinins, and auxin accumulation, and the content of total flavonoids and proanthocyanidins in 30-day-old red pods was significantly higher than that in green pods. This metabolic profile contributes to valuable insights into the metabolic regulation mechanism in different color pods.

## 1. Introduction

*Caragana intermedia* (*C. intermedia*) is a leguminous shrub widely distributed in sandy grasslands and desert regions of west and northwest China and Mongolia [1]. *C. intermedia* is rich in secondary metabolites and natural active substances such as flavonoids. It has good nutritional and medicinal values, and thus is a traditional Chinese medicinal plant used as a tonic. The whole plant, roots, flowers, and seeds can be used in Chinese and Mongolian medicine. Because of its strong tolerance to drought, cold, saline, and other abiotic stresses, *C. intermedia* has attracted much attention in recent years [2]. For example, a dehydration-responsive element-binding (DREB) protein from *Caragana korshinskii* (*C. korshinskii*, one of the closest relatives of *C. intermedia*) is involved in the response to drought, high salt, and low temperature stresses. Transgenic tobacco plants overexpressing *CkDREB* exhibit higher tolerance to salinity and osmotic stresses, and have induced expression of a downstream target gene under normal conditions [3]. High-throughput sequencing of *C. intermedia* miRNAs identified both conserved and novel miRNAs and their target genes in *C. intermedia*. One hundred and forty two miRNAs were identified, thirty eight miRNA targets were predicted, and four targets were validated. The expression of 12 miRNAs in salt-stressed leaves was assessed by qRT-PCR [4]. The metabolic consequences of drought stress were characterized in *C. korshinskii*. Results showed that the abundance of various small carbohydrates and soluble amino acids was increased under drought stress; these compounds may act as compatible solutes or antioxidants [5].

Recent studies have shown that flavonoids, stilbenoids, terpenoids, and other metabolites have strong antioxidant capacity and medicinal usage. A flavonoid named 3,4-dihydroxy-8,9-methylenedioxypterocarpan was extracted from *Caragana jubata* (*C. jubata*). This flavonoid has higher antioxidant activity than vitamin C and butylated hydroxytoluene [6]. Pruinosanone A–C, the anti-inflammatory isoflavones, are derived from roots of *Caragana pruinosa* (*C. pruinosa*) and significantly inhibit the inducible nitric oxide production [7]. Total flavonoids in *Caragana* could protect brain microvascular endothelial cells (BMECs) in humans from hypoxia/reoxygenation (H/R)-induced injury in a dose-dependent manner. Flavonoids can promote angiogenesis in BMECs by activating the HIF-1α-VEGF-Notch 1 signaling pathway [8]. Caragaphenol A selectively inhibits the growth of human gastric cancer cells, and induces cell cycle arrest at G2/M phase and apoptosis with the increased intracellular reactive oxidative species (ROS) level [9]. Hypaphorine, an indole alkaloid isolated from *C. korshinskii*, prevents the differentiation of 3T3-L1 preadipocytes into adipocytes and improves insulin sensitivity in vitro [10]. Carainterol A, a eudesmane sesquiterpenoid extracted from *C. intermedia*, can increase the protein levels of IRS-1, the phosphorylation of the downstream protein kinase AKT, and insulin pathway sensitivity [11]. These secondary metabolites should be the important active ingredients in the genus of *Caragana*.

Many bioactive metabolites from *Caragana* have been isolated and identified with chromatographic or NMR procedures [6,7,10]. However, up to now, no reports link the difference of metabolites and metabolic pathways with different pod colors in *C. intermedia*. The pods of *C. intermedia* show different colors naturally, even at the very early stage of pod development. Is it just a random phenomenon, or the result of evolution and adaptation to environment? Our in-depth analysis revealed the metabolic differences between red and green pods of *C. intermedia*. Based on this, a molecular regulation model of *C. intermedia* pods color formation was proposed. These results also shed light on further study and utilization of plant flavonoids and other metabolites.

## 2. Results

### 2.1. Multivariate Analysis of Metabolites Related to Pod Color and Growth Stage

Multivariate analysis was used to investigate the metabolic differences in different color and different growth stage of red and green pods of *C. intermedia*. The principal component analysis (PCA) and partial least squares-discriminant analysis (PLS-DA) of the metabolite data separated two different color pods regarding metabolite composition and content (Figure 1), reflecting that major differences existed within the red and green pods at metabolite levels. The principal components 1 (PC1) and 2 (PC2) accounted for 41.6% of the variation in separating different pods. In the PLS-DA score plot, PLS1 and PLS2 explained 25.9% and 15.6% of the total variance, respectively. The results demonstrated a clear clustering of different color samples with different growth times.

From the results of PCA and PLS-DA, it was found that there was little difference in metabolites between 10- and 20-day-old pods, but there indeed was difference between 10-and 30-day-old pods. Further analysis showed that the most difference in metabolites existed between 30-day-old red and green pods, so the 30-day-oldsample analysis was carried out in follow-up experiments.

A total of 557 analytes were identified from *C. intermedia* pods (Appendix A), including 348 annotated metabolites. In the approach reported here, 96 distinct metabolites in 15 categories of 30-day-old samples were selected, based on the variable importance in projection (VIP) values (VIP > 1.0) that were obtained from the PLS-DA and fold change (Appendix A). The primary metabolites identified as distinct metabolites in red and green pods mainly consisted of amino acids and nucleotide derivatives. The results reveal that the relative levels of amino acids such as arginine, tyrosine, cysteine, threonine and tryptophan were significantly higher in the red pods than in the green pods. Aromatic amino acids, tryptophan and tyrosine, were the most varied amino acids followed by cysteine, phenylalanine and arginine. We also noticed that the content of different nucleotide derivates, such as guanosine and inosine, did not display a uniform pattern in different pods.

The changes in secondary metabolites in different color pods were also investigated. Twenty three flavonoids (mainly quercetin, apigenin, luteolin, and anthocyanin), 13 phytohormones (mainly zeatin, and indoleacetic acid), eight phenolamines (mainly noradrenaline, *N*-*p*-coumaroylputrescine derivative, and spermidine derivative) and three polyphenols (sinapoyl*O*-hexoside, caffeoyl shikimic acid, and quinic acid) were identified (Appendix A). The significantly different flavonoids (Fold>2 or <0.5, VIP > 1) of 30-day-old green and red pods are shown in Table 1, from which we knew that the red pods showed improved flavonoid metabolism.

### 2.2. Differential Metabolite Profiling in Red and Green Pods of C. intermedia

Analysis of metabolic networks linked to the identified metabolites determined that widely targeted metabolite profiling of 30-day-old green and red pods mainly encompassed in 21 metabolic pathways, in which four had significant difference. As shown in Table 2, results included the metabolic pathways of metabolites found in the Kyoto Encyclopedia of Genes and Genomes (KEGG), which was helpful to analyze the accumulation pattern of metabolites of color formation of pods. As for samples of different growth periods of the same color, the metabolites of 10-day-old and 30-day-old green pods were distributed in 33 metabolic pathways, and there were 10 significantly different pathways. For red samples, there were 32 metabolic pathways and 7 significantly different pathways. The results are shown in Table 2.

The pattern of identified metabolites was analyzed and quantified in all samples. The metabolites appear to be differently accumulated in different growth time and color samples. The results of the analysis of metabolites with distinct differences (VIP > 1) in 30-day-old samples are shown in the form of heat map (Figure 2). The most abundant compounds were metabolites in the shikimic acid pathway and the phytohormone metabolic pathway. The metabolite content in green pods was apparently lower than that in red pods.

Flavonoids are the dominant secondary metabolites. Seventy-four flavonoids were detected in this study, including 46 *O*-glycosides and 9 *C*-glycosides (Appendix A). As expected, twenty three apparently differential flavonoids were detected between green and red pods (Table 1). Several apigenins, luteolins, and anthocyanins were detected in large and varying amounts. The content of apigenin in red pods was significantly higher than that in green pods, and the content of apigenin-7-*O*-glucoside in red pods was 72 times higher than that in green pods. The content of anthocyanin in red pods was higher than that in green pods, although the difference was not significant. Only one quercetin derivative (quercetin-3-*O*-rhamnoside) had significant difference between two color pods, with the content in red pods 25 times higher than that in green pods. However, the content of proanthocyanidins in red pods was apparently higher than that in green pods, and may be converted into colored anthocyanins under suitable conditions (Figure 3). Also notable is chlorogenic acid, which was 5 times higher in green pods than in red pods. Furthermore, sinapoyl *O*-hexoside was also a distinct polyphenol metabolite (VIP > 1, Fold = 2.427).

The metabolites identified as distinct metabolites of the different growth time of red and green pods mainly consisted of phytohormones. The top ten most significantly distinct metabolites of 10-day-old and 30-day-old red and green pods are shown in Table 3. Comparing the distinct metabolites of 10-day-old and 30-day-old different color pods, it was found that phytohormones were the most variable metabolites. There were six and five cytokinins and auxins in the top ten most significantly distinct metabolites in red and green pods respectively. According to existing knowledge about plant phytohormones [12,13], the presence of these phytohormones in pods should promote pod and seed growth, and they could also play other important functions.

### 2.3. Analysis of Metabolite Accumulation Patterns in Red and Green Pods of C. intermedia

The metabolic dataset was assessed in a point-by-point manner. The tentatively identified compounds were assigned into common metabolic pathways. Not surprisingly, the flavonoid metabolic pathway was among the most active pathways in both red and green pods. In recent years, scientists have identified many flavonoids from *Caragana* [6,7,8] and many have been proven to have different biological activities [7,8,11]. To further verify the results in *C. intermedia*, we determined the contents of total flavonoids and proanthocyanidins in 30-day-old pod samples (Figure 3). The results showed that the content of total flavonoids and proanthocyanidins in red pods was significantly higher than that in green pods. Hichriet et al. reviewed the roles of flavonoids in plants [14]. Flavonoids are rich in color, such as apigenin, quercetin, and anthocyanin, which present different colors, especially anthocyanins, which could endowred to blue pigmentation to plants. Proanthocyanidins are oligomeric and polymeric end products of the flavonoid biosynthetic pathway, of which the name reflects the fact that, upon acid hydrolysis, the extension units are converted to colored anthocyanidins [15]. To sum up, red pods have a richer color material base.

Meanwhile, an analysis of the data revealed that phytohormones metabolic pathway were also active. Various metabolites in this pathway, including zeatin and indoleacetic acid, were detected. Overall, the content of phytohormones in red pods was significantly higher than that in green pods. Analyzing the results of others [16,17,18,19,20,21] and ourselves, we suggest that elevated phytohormones might function in promoting the synthesis of anthocyanins.

The levels of metabolites in all samples were compared, and the significantly changed metabolites (VIP > 1) are marked in Figure 4. The notes on metabolic pathways of metabolites were based on the Kyoto Encyclopedia of Genes and Genomes (KEGG) [22].

## 3. Discussion

### 3.1. Flavonoids Are Highly Correlated with Pod Color

Most of the 74 flavonoids detected in red pods were more abundant than those in green pods, suggesting that these flavonoids are up-regulated in red pods. Flavonoids are widely known for their antioxidant ability, coloration to plant tissues, contribution to plant fitness, resistance to biotic and abiotic stresses, and impacts on food quality [23]. As one of the largest groups of secondary metabolites, flavonoids are well recognized in many plants, including *Caragana*. Cui et al. [6] showed that a flavonoid from *C. jubata* had high antioxidant activities to 2,2-diphenyl-1-picrylhydrazyl (DPPH) radicals. Pruinosanone A–C isolated from the roots of *C. pruinosa* was found to significantly inhibit the expression of inducible nitric oxide synthase (iNOS) protein, thus inhibiting nitric oxide (NO) production [7]. Nakabayashi et al. [24] reported that flavonoid accumulation resulted in the improvement of radical scavenging activity in vitro, leading to the enhancement of oxidative and drought tolerance in vivo and prevention of water loss in *Arabidopsis thaliana*. Flavonoids contribute to the organoleptic quality of plant-derived products, and have been shown to be beneficial to human health in the prevention of cell ageing [14]. These data confirmed the usefulness of flavonoids in enhancing both biotic and abiotic stress tolerance in plants.

Anthocyanins, one kind of flavonoid, are pigments responsible for colors in plants, flowers, and fruits. They have been implicated in tolerance to stressorsas diverse as drought, UV-B, and heavy metals, as well as resistance to herbivores and pathogens, efficiently scavenging free radicals and reactive oxygen species [25]. In brief, they are critical for plant survival. In the present study, anthocyanins in red pods of *C. intermedia* were higher than that in green pods. Because of the high content of flavonoids and anthocyanins in red pods, the ability of *C. intermedia* plants producing red pods to adapt to the environment and to tolerate biotic and abiotic stresses might be stronger than those producing green pods. Our current investigation indicates that most *C. intermedia* set red pods. We speculate that this might be a protective measure to adapt to the environment.

### 3.2. Differential Insect Feeding Rate in Red and Green Pods of C. intermedia

What is the practical meaning of having a colorful pod? It is really an interesting question, and one which puzzled us. During the study of color formation mechanism of *C. intermedia* pods, we observed that the insect feeding rate on red pods was lower than that on green pods. We then randomly picked up 30 30-day-old pods from each of the ten green and red pod sampling plants respectively, total 300 pods as one sample, and counted their insect feeding rate. There was no difference between the appearance of insect feeding pods and normal pods. The internal status of normal and insect feeding pods is shown in Figure 5a. It was found that the insect feeding rate of green pods was significantly higher than that of red pods (Figure 5b). Through literature retrieval, we found that there have been reports of the extraction of active substances such as flavonoids from plants to reduce insect feeding activity. Lane et al. [26] found that isoflavonoids from root of *Lupinus angustifolius* showed high feeding-deterrent activity associated with high antifungal activity. Jackowski et al. [27] tested hops flavonoids’ deterrent activity against stored product pests, and the results showed that the pests responded to the tested compounds with considerable reduction of its feeding activity. These results coincide with ours, in that the insect feeding rate of red pods was significantly lowered. Flavonoids such as anthocyanins and proanthocyanidins have function to provide protection against microbial pathogens, insect pests, and larger herbivores [14,15,25]. The contents of flavonoids, anthocyanins, and procyanidins in red pods were up-regulated, which increased their insect-resistance and decreased the insect feeding rate. In following studies, we will systematically study the feeding-deterrent activity of red and green pods of *C. intermedia*, referring to other scholars’ deterrent activity research methods.

### 3.3. Possible Metabolic Regulation Mechanism of Pod Color Variation in C. intermedia

With deep analysis of the metabolomics data, we noticed that the shikimic acid pathway and the phytohormone metabolic pathway were very active in red pods. The content of shikimic acid pathway metabolites in red pods was higher than that in green pods. As the major group of metabolites in this pathway, flavonoids can be converted into proanthocyanidins and anthocyanins, which make the pods red. Therefore, the high content of flavonoids (including proanthocyanidins and anthocyanins) is the direct reason for the color formation in red pods. According to previous studies [14,23,24], we also speculate that the high content of flavonoids in red pods may be closely related to the enhancement of stress tolerance. Therefore, the change in the color of pods is a manifestation of their adaptation to the living environment. In the course of evolution, green pod plants will gradually disappear because of their weak stress tolerance, and only red pod plants will be left.

In addition to the well-known color responsive metabolites such as anthocyanins, we reported here the coordinated response of a number of other metabolites, such as phytohormones, in *C. intermedia*. The phytohormones and their derivatives identified in all analyzed samples, mainly zeatin and indoleacetic acid, have many different categories and the contents change dramatically. Existing evidence indicates that phytohormones are important signaling molecules in regulating plant growth and development throughout the life cycle [28]. There have been many reports on phytohormones promoting the accumulation of anthocyanins. Ozeki et al. [19,20] studied and determined the appropriate concentrations of different auxins and cytokinins to induce anthocyanin synthesis in a carrot suspension culture. Meyer et al. [17] analyzed the effects of auxins and cytokinins on the growth of callus and the yield of anthocyanins produced in vitro by selected callus cultures of *Oxalis linearis*, and found that auxins and cytokinins could promote anthocyanin synthesis in a concentration dependent manner. Thomas et al. [16] demonstrated that 0.5 mg/L of 2,4-d was the best group of many tested phytohormones toenhance anthocyanin production in grape cell cultures. Nerman et al. [18] established callus of *Crataegus sinaica* by culturing stem and leaf explants on MS medium (Murashige and Skoog medium) supplemented with different combinations of phytohormones. A striking increase in anthocyanins was stimulated in stem derived callus containing 2 mg/L BA and 1mg/L NAA. Wang et al. [21] reported that low auxin concentrations were conducive to anthocyanin accumulation in red-fleshed apple calli, and that the changes in the anthocyanin content of calli grown under different auxin conditions were also reflected at the transcript level. These reports suggest that phytohormones participate in the regulation of anthocyanin accumulation, presumably via multiple metabolic regulation pathways.

To sum up, different pod colors of *C. intermedia* are caused by the accumulation of different metabolites, which may be a selection for *C. intermedia* to adapt to the living environments. Based on the knowledge of metabolic profiling in different color pods, the mechanism of metabolic regulation is proposed, providing new materials for the study of plant stress tolerance.

## 4. Materials and Methods

### 4.1. Plant Materials

The examined red and green pods were collected from wild *C. intermedia* in Liangcheng County, Inner Mongolia, China in June 2016. Samples of different colors with the same growing time were collected at the same time. Representative red and green pods were shown in Figure 6. We collected 5-, 10-, 15-, 20-, 25-, and 30-day-old pods. The preliminary results showed that there were differences between 10-, 20-, and 30-day-old pods, so the metabolomic analysis was conducted with these three time point samples. Ten red pod plants and ten green pod plants were selected for sampling in this experiment. All samples were labeled before pod setting, and the selected plants grew under the same conditions. When sampling, the beans were removed and only the pod skins were collected. There were three biological replicates of both red and green pods, each with 20 pods randomly collected from ten selected plants. All samples were snap-frozen in liquid nitrogen for metabolite extraction.

### 4.2. Sample Preparation and Extraction

The freeze-dried pods were crushed using a mixer mill (MM400, Retsch, Haan, Germany) with a zirconia bead for 1.5 min at 30 Hz. For each powdered sample,100 mg was used for extraction overnight at 4 °C with 1.0 mL 70% aqueous methanol. Following centrifugation at 10,000× *g* for 10 min, the extracts in the supernatant were passed through the SPE cartridge (CNWBOND carbon-GCB, 250 mg, 3 mL; ANPEL, Shanghai, China) and filtered (SCAA-104, 0.22 μm pore size; ANPEL, Shanghai, China) before LC-ESI-MS/MS analysis.

### 4.3. High Performance Liquid Chromatography (HPLC) Conditions

The sample extracts were analyzed using a LC-ESI-MS/MS system (HPLC, Shim-pack UFLC SHIMADZU CBM30A system, Kyoto, Japan; MS, Applied Biosystems 4500 QTRAP, Foster city, CA, USA). The analytical conditions were as follows: HPLC: column, Waters ACQUITY UPLC HSS T3 C18 (1.8 µm, 2.1 mm × 100 mm; Milford, MA, USA); solvent system, solvent A (water, 0.04% acetic acid): solvent B (acetonitrile, 0.04% acetic acid); gradient program, 100:0 (*v*/*v*) at 0 min, 5:95 (*v*/*v*) at 11.0 min, 5:95 (*v*/*v*) at 12.0 min, 95:5 (*v*/*v*) at 12.1 min, 95:5 (*v*/*v*) at 15.0 min; flow rate, 0.40 mL·min^−1^; temperature, 40 °C; injection volume: 5 μL. The effluent was alternatively connected to an ESI-triple quadrupole-linear ion trap (Q Trap)-MS.

Quality control (QC) samples were prepared from the mixture of all samples, then divided into four samples and analyzed under the same conditions as the experimental samples. The QC samples were injected every six experimental samples throughout the analytical procedure, to provide a set of data from which repeatability could be assessed.

### 4.4. ESI-Q TRAP-MS/MS

The LC-ESI-MS/MS was performed by Metware Biotechnology Co., Ltd. (Wuhan, China). Linear ion trap (LIT) and triple quadrupole (QQQ) scans were acquired on a triple QTrap, API 4500 QTrap LC/MS/MS system (Foster city, CA, USA), equipped with an ESI Turbo Ion-Spray interface, operating in positive ion mode and controlled by Analyst 1.6 software (AB Sciex, Redwood City, CA, USA). The ESI source operation parameters were as follows: ion source, turbo spray; source temperature 550 °C; ion spray voltage (IS) 5500 V; ion source gas I (GSI), gas II (GSII), curtain gas (CUR) were set at 55, 60, and 25.0 psi, respectively; the collision gas (CAD) was high. Instrument tuning and mass calibration were performed with 10 and 100 μmol·L^−1^ polypropylene glycol solutions in QQQ and LIT modes, respectively. QQQ scans were acquired as multiple reaction monitoring (MRM) mode experiments with the collision gas (nitrogen) set to 5 psi. De-clustering potential (DP) and collision energy (CE) for individual MRM transitions were performed with further DP and CE optimization. A specific set of MRM transitions were monitored for each period, according to the metabolites eluted within this period.

### 4.5. Metabolite Identification and Quantification

Metabolite identification was based on the primary and secondary spectral data annotated against public databases, namely MassBank [29], KNAPSAcK [30], HMDB [31], MoToDB [32], and METLIN [33]; a self-compiled database named MWDB (MetWare biological science and Technology Co., Ltd. Wuhan, China), following the standard metabolic operating procedures [34].

### 4.6. Statistical Analysis

Metabolite quantification was carried out using principle component analysis (PCA) and partial least squares discriminant analysis (PLS-DA) to study metabolite color-specific accumulation. Metabolites with significant differences in content were set with variable importance in projection (VIP) and fold change between different color pods.

To further declare the biological significance associated with pod color, a metabolic pathway that links differential metabolites to metabolic pathways was constructed according to the Kyoto Encyclopedia of Genes and Genomes (KEGG) database.

### 4.7. Determination of Total Flavonoids

Extraction conditions of total flavonoids were as follows: 70% aqueous methanol was the solvent; material to solvent ratio was 1:20; temperature of ultrasonic extraction was 60 °C, power 160 W, time 70 min; material was immersed for 24 h after ultrasonic treatment. Total flavonoid contents of the pod extracts were determined as described by Chukwumah [35]. Briefly, 0.08 mL of 5% sodium nitrite was added to 0.8 mL of sample extract. After 6 min, 0.08 mL of 10% aluminum chloride was added. 0.8 mL of 4% sodium hydroxide and 0.24 mL of distilled water were added to the mixture after 6 min. After 15 min, the sample extract was centrifuged for 10 min at 8000 rpm, and the absorbance of the supernatant was read at 510 nm using a spectrophotometer (GEN10S UV-Vis, www.thermofisher.com). A rutin standard curve (0–60 μg/mL) was established and used to calculate the total flavonoids content of the sample extracts (Appendix A).

### 4.8. Determination of Proanthocyanidin

The total proanthocyanidin content in the pod extracts was determined by vanillin–hydrochloric acid method assay [36]. The ultrasonic extraction conditions were: material to solvent ratio 1:25, temperature 40 °C, power 160 W, time 70 min. After ultrasonic treatment, the sample extract was centrifuged for 10 min at 5000 rpm, then 0.2 mL of supernatant was placed in a tube into which 1.2 mL of methanol solution containing 4% vanillin was added. After vigorous stirring, 0.6 mL of concentrated hydrochloric acid was added to the solution. After stirring, the reaction was set in a 30 °C water bath in the dark for 30 min. The absorbance of the solution was measured at 500 nm using a spectrophotometer. A standard curve with proanthocyanidin (0–100 μg/mL) was used to calculate the amount of proanthocyanidin in the sample extracts (Appendix A).

## 5. Conclusions

The metabolites of red and green pods of *C. intermedia* were analyzed by broadly targeted metabolomics. The results showed that phytohormones, flavonoids, and anthocyanins were increased and the insect feeding rate was decreased in red pods. Taken together, our results indicate that the shikimic acid pathway and the phytohormone metabolic pathway closely contribute to the formation of pod color of *C. intermedia*. With the increase of anthocyanin and other flavonoids in red pods, the biotic and abiotic adaptations of pods of *C. intermedia*, like the insect feeding rate, were improved. This study shows the potential of flavonoid content modifications to alter the adaptability of plants to their living environment.

## Figures and Tables

**Figure 1 molecules-24-00717-f001:**
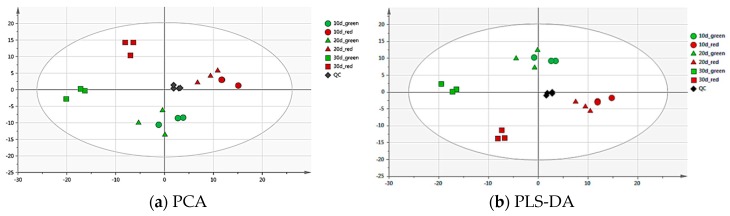
Principal component analysis (PCA) (**a**) and partial least squares-discriminant analysis (PLS-DA) (**b**) score plot. Score plots were derived using LC-ESI-MS/MS datasets from the red and green pod samples. The X axis represents PC1, and the Y axis represents PC2. Each sample has three biological duplicates, and is been represented on the plot by a unique symbol: the green circle, 10-day-old green pods; the red circle, 10-day-old red pods; the green triangle, 20-day-old green pods; the red triangle, 20-day-old red pods; the green square, 30-day-old green pods; the red square, 30-day-old red pods; the black diamond, QC.

**Figure 2 molecules-24-00717-f002:**
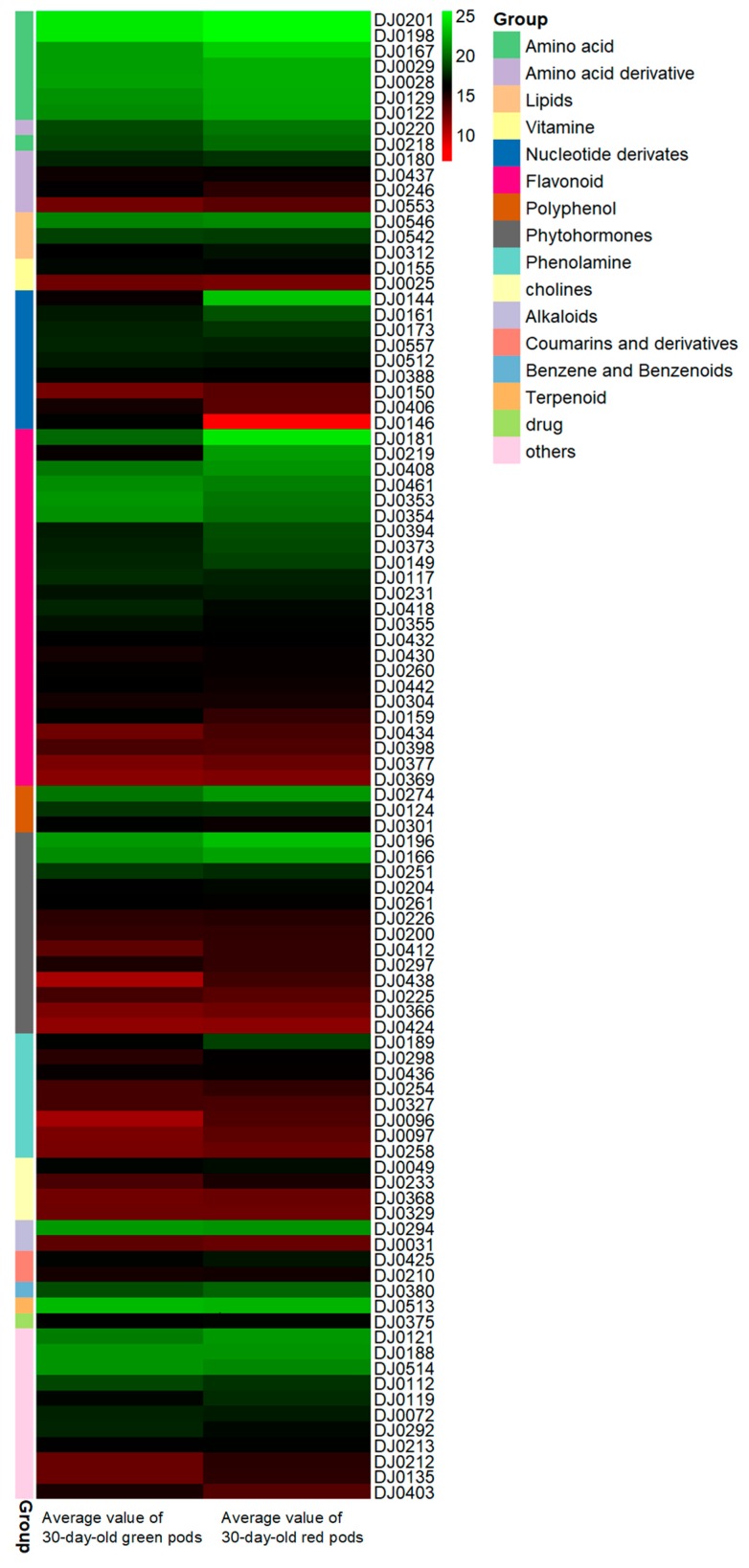
Significantly different metabolites in 30-day-old green (**left**) and red (**right**) pods. Significantly different metabolites of 30-day-old green and red pods after anthesis were assessed by variable important in projection (VIP) and fold change. Each different color rectangle on the left side of the heatmap represents one metabolite. The content of different metabolites in each class decreases from top to bottom in the samples. The numbers on the right side of the picture were the codes of the result of the LC-ESI-MS/MS, with specific information shown in Appendix A. The names of different types of metabolites are displayed on the right side of the chart. The values in picture are the averages of three biological duplicate samples.

**Figure 3 molecules-24-00717-f003:**
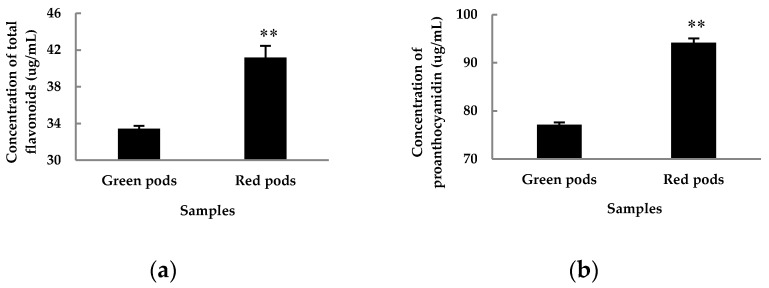
Concentration of total flavonoids (**a**) and proanthocyanidin (**b**) in 30-day-old red and green pods after anthesis. The values are the averages of three biological duplicate samples. ** indicates significant difference between samples at *p* < 0.01 level using Student’s *t*-test.

**Figure 4 molecules-24-00717-f004:**
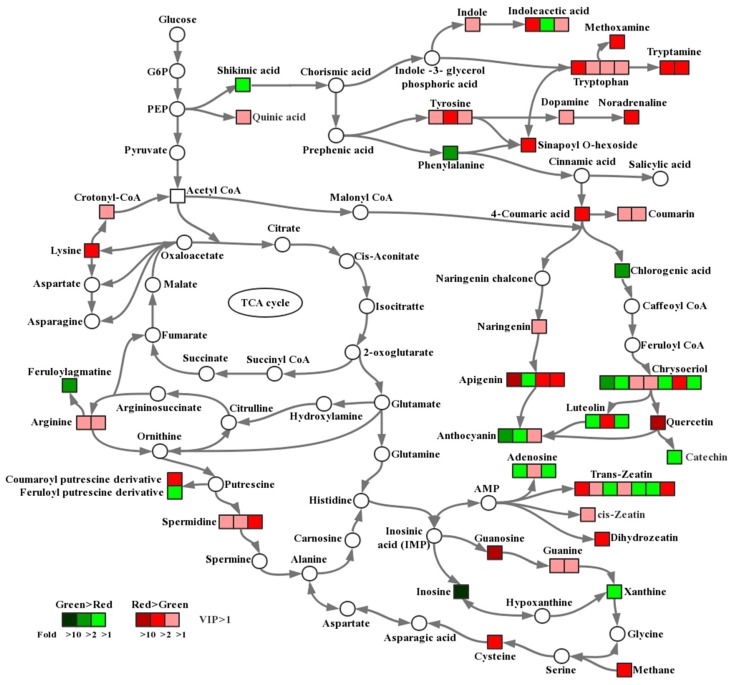
Comparison of metabolite accumulation in different pathways between green and red pods. Significant metabolite changes (VIP > 1) observed in red pods versus green pods are shown in the metabolic map. Metabolites shown as squares were detected in this study, whereas those shown as circles were undetectable. The squares in red indicate that the content of the metabolite in red pods was higher than that in green pods, and the darker the color, the larger the difference. Similarly, the squares in green indicate that the content of the metabolite in green pods was higher than that in red pods, and the darker the color, the larger the difference. The level of significance was set at VIP > 1.

**Figure 5 molecules-24-00717-f005:**
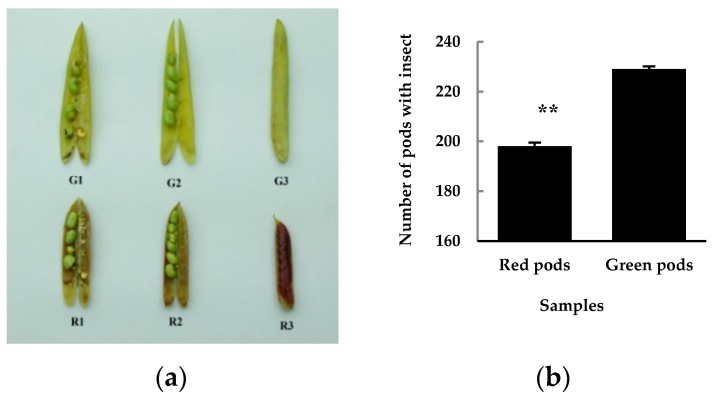
The insect feeding 30-day-old green (**top**) and red (**bottom**) pods after anthesis (**a**), and their insect feeding rates (**b**). The samples in A are: G1, R1: insect feeding pods; G2, R2: normal pods (inside) and G3, R3: normal pods (intact). The value in B is the average of three biological duplicate samples.

**Figure 6 molecules-24-00717-f006:**
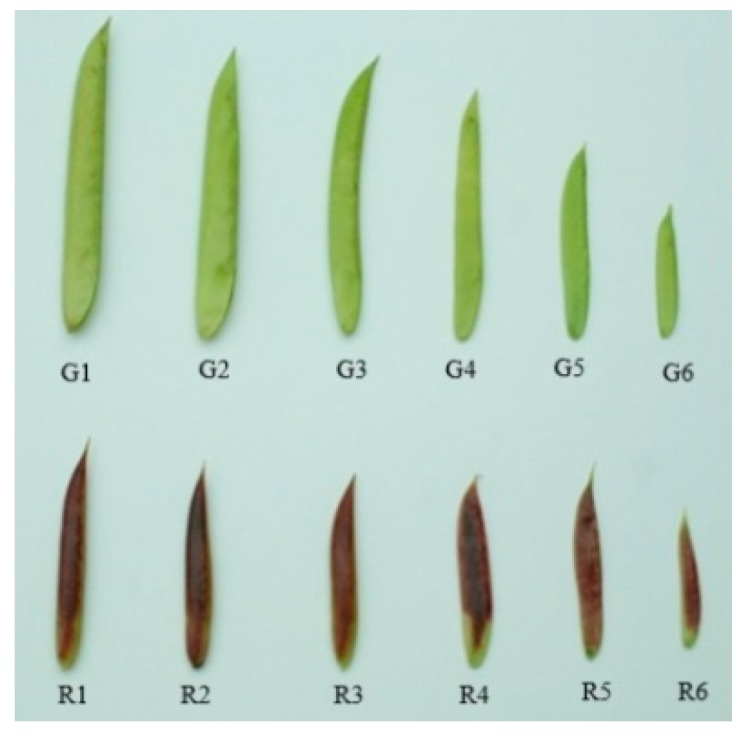
Representative samples of green (**top**) and red (**bottom**) pods. The sample pods are as follows: G6, R6: 5-day-old pods; G5, R5: 10-day-old pods; G4, R4: 15-day-old pods; G3, R3: 20-day-old pods; G2, R2: 25-day-oldpods and G1, R1: 30-day-old pods after anthesis.

**Table 1 molecules-24-00717-t001:** Significantly different flavonoids of 30-day-old green and red pods.

Index	Component Name	Peak Area of Green Pods	Peak Area of Red Pods	Fold (Red/Green)	VIP ^1^
DJ0219	Apigenin-7-*O*-glucoside	49,167	3,540,000	72.000	2.42218
DJ0181	Quercetin-3-*O*-rhamnoside	970,667	24,600,000	25.343	2.03162
DJ0394	Luteolin *O*-malonylhexoside	154,000	507,000	3.292	1.31471
DJ0373	Apigenin 7-*O*-glucoside	174,333	451,333	2.589	1.24127
DJ0434	Selgin*O*-hexoside derivative	5987	14,143	2.362	1.02893
DJ0408	Apigenin *O*-malonylhexoside	1,373,333	2,820,333	2.054	1.03458
DJ0353	Peonidin *O*-hexoside	2,923,333	1,323,333	0.453	1.00394
DJ0354	Chrysoeriol 5-*O*-hexoside	2,586,667	1,136,667	0.439	1.04638
DJ0159	Chlorogenic acid	75,300	20,800	0.276	1.1813

^1^ VIP is variable importance in projection.

**Table 2 molecules-24-00717-t002:** Significantly different metabolic pathways ^1^ of different pods growth stages (10- and 30-day-old) and different color pods (30-day-old green and red pods).

Metabolic Pathway	Total Number of Metabolites	Detected Metabolites in Green Pods (10- and 30-Day-Old) (*p* Value)	Detected Metabolites in Red Pods (10- and 30-Day-Old) (*p* Value)	Detected Metabolites in 30-Day-OldGreen and Red Pods (*p* Value)
Aminoacyl-tRNA biosynthesis	75	7 (0.00002)	-	-
ABC transporters	90	6 (0.0005)	-	4 (0.006)
Cyanoamino acid metabolism	41	5 (0.00008)	4 (0.001)	-
Cysteine and methionine metabolism	56	5 (0.0004)	4 (0.004)	-
Purine metabolism	92	4 (0.02)	7 (0.00009)	3 (0.04)
Tyrosine metabolism	76	4 (0.01)	-	3 (0.02)
Sulfur metabolism	18	2 (0.02)	2 (0.02)	-
Sphingolipid metabolism	25	2 (0.03)	2 (0.04)	-
Phenylalanine, tyrosine and Tryptophan biosynthesis	27	2 (0.04)	2 (0.04)	-
Selenoamino acid metabolism	30	2 (0.046)	-	-
Thiamine metabolism	26	-	2 (0.04)	-
β-Alanine metabolism	31	-	-	2 (0.03)

^1^ Significantly different metabolic pathways are distinguished by *p* Value. When *p* > 0.05, detected metabolites are represented by “-”.

**Table 3 molecules-24-00717-t003:** Distinct metabolites of red and green pods at different growth stages.

Component Name	Class	Peak Area of 10-Day-Old Red Pods	Peak Area of 30-Day-OldRed Pods	Peak Area of 10-Day-OldGreen Pods	Peak Area of 30-Day-OldGreen Pods	Fold Change ^1^ (Red/Green)	VIP (Red/Green)
Inosine	Nucleotide derivates	2.53 × 10^5^	1.00 × 10^2^	--	--	2533.67/--	3.17/--
iPR	Nucleotide derivates	2.96 × 10^6^	9.33 × 10^3^	8.08 × 10^5^	3.90 × 10^4^	316.79/20.69	3.90/3.04
tZR	Phytohormones	8.13 × 10^6^	2.18 × 10^5^	5.69 × 10^6^	2.97 × 10^5^	37.31/19.15	3.05/2.78
iP	Phytohormones	5.86 × 10^4^	6.10 × 10^3^	2.78 × 10^4^	4.55 × 10^3^	9.60/6.12	2.42/2.17
IAA	Phytohormones	3.06 × 10^4^	3.34 × 10^3^	3.16 × 10^4^	3.02 × 10^3^	9.14/10.45	2.38/2.47
trans-Zeatin riboside-*O*-glucoside	Phytohormones	7.54 × 10^4^	9.70 × 10^3^	1.14 × 10^5^	1.48 × 10^4^	7.78/7.72	2.32/2.30
DZR	Phytohormones	3.38 × 10^5^	5.69 × 10^4^	--	--	5.94/--	2.14/--
tZROG	Phytohormones	1.46 × 10^5^	2.64 × 10^4^	1.83 × 10^5^	2.39 × 10^4^	5.53/7.68	2.09/2.29
Caffeoyl shikimic acid	Polyphenol	1.94 × 10^5^	4.46 × 10^4^	--	--	4.36/--	1.94/--
LPC (1-acyl 18:1)	Lipids	1.24 × 10^6^	3.25 × 10^5^	--	--	3.83/--	1.15/--
Guanosine	Nucleotide derivates	--	--	8.06 × 10^6^	4.74 × 10^4^	--/169.95	--/2.04
Crotonoside	others	--	--	3.62 × 10^4^	5.67 × 10^2^	--/63.94	--/2.15
Spermidine derivative	Phenolamine	--	--	1.38 × 10^5^	2.62 × 10^4^	--/5.28	--/1.75
Dopamine	others	--	--	1.42 × 10^4^	3.24 × 10^3^	--/4.38	--/2.09

^1^ Fold change is peak area of 30-day-old/peak area of 10-day-old. The value before “/” is the red pods, after “/” is the green pods.

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
