# Peer review of "Analysis of Metabolite Accumulation Related to Pod Color Variation of Caragana intermedia"

_molecules, 2019, doi:10.3390/molecules24040717_

Round 1
Reviewer 1 Report
The article « Analysis of the Metabolites Accumulation Related to Pods Color Variation of Caragana intermedia» was modified in the revision process. The authors have re-worked the description of the analytical part of the manuscript, but they failed to clarify it. It seems doubtful, that the analysts, who had actually performed the measurements were able to revise the text of this article. Main issue is the description of the HPLC-MS method:
Line 326. The multiple reaction monitoring (MRM) was performed by Metware Biotechnology Co., Ltd. (Wuhan, China). – MRM is a detection mode, not the data handling technique.
Line 335. De-clustering potential (DP) and collision energy (CE) for individual MRM transitions were performed with further DP and CE optimization – Such optimization may be conducted by using standards of individual compounds (giving individual MRM transitions). Such standards were not employed according to the text of the manuscript.
Line 336. The MRM for each sample was performed in triplicate [28]. – Not clear what does this mean. Was each sample analyzed in triplicate by HPLC-MS in MRM mode? What shows this reference?
Line 345-347. Excessive and misleading explanation of the MRM regime.
Line 348-350. “the peak area of the mass spectra of all substances was integrated, and the mass spectra of the same metabolites in different samples were corrected” Please re-phrase this and explain how the normalization was performed for peak areas or for mass-spectra? And what mass-spectra are implied in MRM mode?
Additionally:
Line 321-324. So QC samples were used only for repeatability? Not for normalization? And a QC sample in this case is sort of the averaged sample prepared of the mixture of all samples?
Author Response
1. Line 326. The multiple reaction monitoring (MRM) was performed by Metware Biotechnology Co., Ltd. (Wuhan, China). – MRM is a detection mode, not the data handling technique.
--Yes, we use LC-ESI-MS/MS to detect, which has been revised in the manuscript accordingly.
2. Line 335. De-clustering potential (DP) and collision energy (CE) for individual MRM transitions were performed with further DP and CE optimization – Such optimization may be conducted by using standards of individual compounds (giving individual MRM transitions). Such standards were not employed according to the text of the manuscript.
--When we analyzed the metabolites, the corresponding substances were optimized in the biological samples, and no standard materials were used to optimize.
3. Line 336. The MRM for each sample was performed in triplicate [28]. – Not clear what does this mean. Was each sample analyzed in triplicate by HPLC-MS in MRM mode? What shows this reference?
-- Sorry for the confusion. Actually, three biological replicates of each samples was analyzed. This was described already in Section 4.1, so this sentence was deleted here. The method used in this reference is the same as what we used. After discussion, we had deleted it because the experimental method had been clearly stated in our manuscript.
4. Line 345-347. Excessive and misleading explanation of the MRM regime.
-- Thank you for your kind suggestion, and we are also aware that this section is too much of a methodological discussion. We had deleted the description about MRM in the manuscript.
5. Line 348-350. “the peak area of the mass spectra of all substances was integrated, and the mass spectra of the same metabolites in different samples were corrected” Please re-phrase this and explain how the normalization was performed for peak areas or for mass-spectra? And what mass-spectra are implied in MRM mode?
--In this experiment, the peak area of the mass spectra is the relative content of the substance, without normalization.
6. Additionally: Line 321-324. So QC samples were used only for repeatability? Not for normalization? And a QC sample in this case is sort of the averaged sample prepared of the mixture of all samples?
-- Yes, QC samples were only used for repeatability, not for normalization. And a QC sample in this manuscript is the averaged sample prepared of the mixture of all samples.

Reviewer 2 Report
Unfortunately, I do not see much improvement in this revised manuscript except some technical points. I still think the authors are required to expand section 3.3 to include their own answer to the question I presented below.
14 among individual pods or plants?
22 “extraordinarily active” in which pod?
23 accumulation, and the…… and was missing?
67 “Is it just a random phenomenon or the result of evolution and adaptation to environment?” Is this question answered? I think not. Please try to give certain explanation on this statement. You provided us with your observation, but not explanation to this question.
88 “it was found that there was little difference in metabolites between 10- and 20-day-old pods, but there indeed was difference between 10-, 20- and 30-day-old pods.” This is very contradictory statement within. “Little difference” vs “indeed was difference.”
Table 1. Define VIP.
116 “Regulation mechanism” stated here has not been described in this manuscript. In 3.3, you only showed up-regulation of anthocyanin and phytohormone metabolisms in red pod. However, what we actually need is what causes upregulation happening.
Tables 2 and 3. Are the metabolisms up-regulated or down-regulated. You stated just “significantly different.” Better combine these two tables.
Figure 2. Small font size would make it almost impossible to read when the heat diagram is reduced for publication.
Figures 3 and 4. How the column height was determined? I wonder how did you quantified flavonoids and proanthocyanidins without using standard compound. You can compare peak areas for one compound but cannot directly compare the quantities between different compounds.
Figure 4. Could you explain what are those multiple boxes for single metabolite represent?
Section 3.3 Here, you provided good explanation on the adaptive force of C. intermdedia in developing anthocyanin accumulation to resist against herbivory. Still explanation further needed is how one pod mobilize anthocyanin biosynthesis while its neighbor does not. Without answering to this fundamental question, this work remains only as preliminary. If you do not have answer to this question, it would be advisable to state your own speculation on this point at least.
Author Response
1. 14 among individual pods or plants?
-- It’s among individual plants, which has been revised in the manuscript.
2. 22 “extraordinarily active” in which pod?
-- We found that it is “extraordinarily active” in red pods, which has been revised in the manuscript.
3. 23 accumulation, and the…… and was missing?
-- Corrected.
4. 67 “Is it just a random phenomenon or the result of evolution and adaptation to environment?” Is this question answered? I think not. Please try to give certain explanation on this statement. You provided us with your observation, but not explanation to this question.
-- It should be the result of evolution. This question had been addressed in the last two sentences of the last paragraph of Section 3.1 and the first sentence of the last paragraph of Section 3.3, which displayed in red color.
5. 88 “it was found that there was little difference in metabolites between 10- and 20-day-old pods, but there indeed was difference between 10-, 20- and 30-day-old pods.” This is very contradictory statement within. “Little difference” vs “indeed was difference.”
--Sorry for the confusion: It was found that there was little difference in metabolites between 10- and 20-day-old pods, but there indeed was difference between 10- and 30-day-old pods. It has been revised in the manuscript accordingly.
6. Table 1. Define VIP.
-- Footer was added for VIP in the manuscript below Table 1.
7. 116 “Regulation mechanism” stated here has not been described in this manuscript. In 3.3, you only showed up-regulation of anthocyanin and phytohormone metabolisms in red pod. However, what we actually need is what causes upregulation happening.
--Based on our research results and others, we speculate that the increase of phytohormone can promote the accumulation of flavonoids and anthocyanins in red pods, but it is not clear that what kind of mechanism promotes the increase of phytohormone. Further study of its regulation mechanism will be the focus of our follow-up research. We intend to study the mechanism of up-regulation of metabolites in red pods such as phytohormone, flavonoids and anthocyanins from the molecular level. It has been revised in the manuscript accordingly.
8. Tables 2 and 3. Are the metabolisms up-regulated or down-regulated. You stated just “significantly different.” Better combine these two tables.
-- Here we list all up- and down-regulated metabolic pathways for enrichment analysis. Tables 2 and 3 have been combined into one table according to your kind suggestion.
9. Figure 2. Small font size would make it almost impossible to read when the heat diagram is reduced for publication.
-- The font in Figure 2 has been enlarged, and the smallest is 24px.
10. Figures 3 and 4. How the column height was determined? I wonder how did you quantified flavonoids and proanthocyanidins without using standard compound. You can compare peak areas for one compound but cannot directly compare the quantities between different compounds.
-- We used standard curve to calculate the content of total flavonoids and proanthocyanidins. We described the standard curve in Section 4.7 and 4.8. And we uploaded the standard curves as supplemental Table S3 in this version of the manuscript.
11. Figure 4. Could you explain what are those multiple boxes for single metabolite represent?
-- Each box represents a detected metabolite. Multiple boxes represent that we detected several derivatives of one metabolite.
12. Section 3.3 Here, you provided good explanation on the adaptive force of C. intermdedia in developing anthocyanin accumulation to resist against herbivory. Still explanation further needed is how one pod mobilize anthocyanin biosynthesis while its neighbor does not. Without answering to this fundamental question, this work remains only as preliminary. If you do not have answer to this question, it would be advisable to state your own speculation on this point at least.
-- After we noticed the different herbivory resist phenotype between the red and green pods, we did hope to explore the genetic and revolution force lay behind. However, it is hard that it takes at least 4 years to set seeds for a seed-grown seedling of C. intermedia, so we need more time to trace down several generations to confirm its genetic stabilities. It has been added in the first paragraph of Section 3.3 according to your kind suggestion, which displayed in red.

This manuscript is a resubmission of an earlier submission. The following is a list of the peer review reports and author responses from that submission.
Round 1
Reviewer 1 Report
In this manuscript, authors have noticed color difference in the pod of C. intermedia. They performed metabolomics analysis to find out metabolic difference between pods with different colors. The red color was found presumably due to flavonoid as well as anthocyanin accumulation. Based on these observation, they proposed “metabolic regulation.”
However, overall the conclusion is too predictable and no new insight into colorant accumulation is proposed. I suspect the “metaboli regulation” they proposed was rather common casual observation without providing any any new mechanism. It is easily conceived that the color might be derived from anthocyanin. Then surely one can expect active metabolism in anthocyanin pathway, starting from shikimate pathway proceeding through phenylalanine biosynthesis and flavonoid synthesis. Therefore, this reviewer judges little new finding.
One minor, but representing total scientific quality of this manuscript is that, in Tables 1 and 4, one cannot see any unit on the column labelled as “content.” Probably what they are claiming to be “content” is, I guess, relative intensity of ion flux or something. Authors have to be more strict on chemistry if they are dealing with metabolomics.
Reviewer 2 Report
The article «Analysis of the Metabolic Regulation Mechanism of Pods Color Variation of Caragana intermedia Determined by a Broadly Targeted Metabolomics» is a good example of the metabolism study based on the highly informative HPLC-MS data. The authors have not only successfully distinguished pod samples of different age and color, but have also explained these differences from the physiological point of view. Found metabolites were classified into several groups depending on their chemical role and participation in the metabolic pathways. However many questions have arisen from the insufficient description of the analytical part of the work.
1) In the line 65 the authors note that many metabolites were isolated and identified in this plant, while not giving a single reference on the studies describing the analytical techniques employed for this.
2) In tables 1,4 and 5 the contents of the metabolites are given as numerical values, probably as peak areas, because standard substances were not employed for quantification. This should be clarified, was any normalization applied for these data? Was the repeatability of the extraction process verified or only biological replicates were analyzed?
3) Line 319-321. Should be rephrased. Please clarify whether SCAN or MRM mode was applied for quantification. In case of MRM mode an optimization for DP and CE is often made by means of the analysis of a mixture of standards in varying conditions.
4) In the line 165 it is stated that all metabolites were “tentatively identified”. This should also be explained, because low resolution QQQ (Q-Trap) instrument was employed. In the paragraph 4.5 the authors declare the use of their own database, what data was included there from these or earlier performed experiments? What “standard metabolic operating procedures” are supposed (since no reference given)? Were the standards of such metabolites used to construct this database?
5) Line 309-310. Please explain further use of QC samples and their role in the quantification made for analyzed pod extracts.
As a suggestion, tables 2 and 3 may be united somehow in a single table, same for tables 4 and 5. Secondly, I recommend the removal of extra web links for the commonly used instruments and materials. MS medium (line 267) may be rephrased to avoid misinterpretation with the MS standing for mass spectrometry. Lastly, conclusion seems rather short and some relevant point from the discussion section may be added.